# Discovery of novel hepatocyte eQTLs in African Americans

Yizhen Zhong, Tanima De , Cristina Alarcon, C. Sehwan Park, Bianca Lec, Minoli A. Perera*

Department of Pharmacology, Northwestern University Feinberg School of Medicine, Chicago, Illinois, United States of America

* minoli.perera@northwestern.edu

## Abstract

African Americans (AAs) are disproportionately affected by metabolic diseases and adverse drug events, with limited publicly available genomic and transcriptomic data to advance the knowledge of the molecular underpinnings or genetic associations to these diseases or drug response phenotypes. To fill this gap, we obtained 60 primary hepatocyte cultures from AA liver donors for genome-wide mapping of expression quantitative trait loci (eQTL) using LAMatrix. We identified 277 eGenes and 19,770 eQTLs, of which 67 eGenes and 7,415 eQTLs are not observed in the Genotype-Tissue Expression Project (GTEx) liver eQTL analysis. Of the eGenes found in GTEx only 25 share the same lead eQTL. These AA-specific eQTLs are less correlated to GTEx eQTLs. in effect sizes and have larger Fst values compared to eQTLs found in both cohorts (overlapping eQTLs). We assessed the overlap between GWAS variants and their tagging variants with AA hepatocyte eQTLs and demonstrated that AA hepatocyte eQTLs can decrease the number of potential causal variants at GWAS loci. Additionally, we identified 75,002 exon QTLs of which 48.8% are not eQTLs in AA hepatocytes. Our analysis provides the first comprehensive characterization of AA hepatocyte eQTLs and highlights the unique discoveries that are made possible due to the increased genetic diversity within the African ancestry genome.

## Author summary

Precision medicine has enabled more accurate diagnoses, treatments and outcomes with the implementation ongoing at many US hospitals. However, much of these efforts are based on genetic associations conducted in European cohorts. The discovery of disease associated genetic variants is lacking in minority populations, leading to a growing disparity in precision medicine. Our study is the first to map eQTLs in hepatocytes obtained exclusively from African Americans. The liver is a clinically relevant organ linked to many diseases, including cardiovascular and lipid traits, in addition to its importance in drug metabolism. Our analysis replicated previously identified eQTLs within the GTEx and uncovered novel regulatory variants. Many of these novel eQTLs were discovered due to the higher allele frequency of these variants in African ancestry populations, thereby demonstrating the potential increase in power with the use of African ancestry genomes. We

**Data Availability Statement:** All relevant data are within the manuscript and its Supporting Information files except for the genotyping and expression files. These are available from the GEO (GSE124076 and GSE147628).

**Funding:** This work was supported by National Institute of Health (NIH)/National Institute on Minority Health and Health Disparities (NIMHD) grant R01 MD009217 and U54 MD010723. The funders had no role in study design, data collection and analysis, decision to publish, or preparation of the manuscript.

**Competing interests:** The authors have declared that no competing interests exist.

decreased the number of potential causal variants at GWAS loci that overlap with our eQTLs. This is important as fine-mapping of causal variants has primarily relied on functional validation. Thus, narrowing the number of potential causal variants expedites this process. Lastly, we show that by identifying both population-specific and hepatocyte-specific eQTLs in African Americans, we can identify potential genes that drive disease.

## Introduction

Many diseases such as coagulopathies and metabolic disorders show disparities in morbidity, mortality and response to drug therapy between populations of African descent (e.g. African Americans) and populations of European descent [1]. For example, the adjusted mean weekly dose of a common anticoagulant drug, warfarin, is higher for AAs (43 mg) than those of European descent (36 mg) [2]. Venous thromboembolism (VTE), a condition in which blood clots form within veins, is a leading cause of death and disability. The incidence of VTE is 30% to 60% higher in AAs than other ethnicities [3]. Recent Genome-Wide Association Studies (GWAS) have been successful in identifying genetic variants that explain the ancestry-related differences in disease susceptibility and drug therapy outcome [4, 5] and have led to recommendations for effective and specialized treatments [6]. However, as the associated variants are mostly within non-coding regions, we still know little about the molecular mechanisms underlying population-differentiated phenotypes.

With the advent of integrative methods to associate gene expression with genetic variation, we have been able to add greater understating to genetically regulated gene expression and the role it plays in complex diseases [7]. eQTL mapping has been conducted in different tissue types [8, 9] and treatment conditions [10, 11] with the recognition of the important role of context in investigating the genetic regulation of gene expression and in pinpointing the causal tissue for complex diseases. Liver is a critical organ for understanding drug metabolism, as well as coagulopathies and metabolic disorders, and is likely to be the causal tissue for many GWAS studies [12–14]. However, eQTL mapping in liver tissue has been primarily conducted in European and Asian populations [9, 15, 16]. Due to the difficulty in obtaining liver from African American (AA) donors, there are no studies performed exclusively in the liver of AAs, leaving the genome-wide regulation of gene expression in populations with high levels of African ancestry unexplored.

Genetic architecture of gene expression varies across populations [17]. eQTL mapping in diverse populations has been shown to implicate novel trait associations reflecting the epidemiological history [18] and adaption of innate immune responses to infection [19]. The catalog of regulatory variants needs to be expanded to include admixed populations such as AAs in disease relevant tissues, as these populations may suffer disproportionally of many chronic disease [20, 21]. Unfortunately, eQTL mapping in AAs has thus far been limited to lymphoblastoid cell lines (LCLs), primary macrophages, adipose and muscle tissues [19, 22–25]. The Genotype-Tissue Expression Project (GTEx) Consortium has added to our understanding of the eQTL landscape using a multiethnic cohort. However, even in this large effort, only 15 individuals of African ancestry were included in the liver eQTL mapping (v7) [26], making extrapolation into this population under-powered. Moreover, GTEx eQTLs were mapped in liver tissue and hence represent the transcriptome of multiple cell types. Therefore, the possibility of missing important population-specific regulatory variants in hepatocytes is a valid concern. In addition, the characteristics of population-specific eQTLs and the degree to which population-specific eQTLs can inform mechanistic understanding of disparities in disease

incidence and outcomes are unknown. AAs are an admixed population with a large portion of their genomes inherited from African populations and a smaller contribution from European populations [27]. As such, AAs have shorter stretches of linkage disequilibrium and greater amount of genetic variation with differing allele frequencies, which could be leveraged to uncover novel regulatory variants in hepatocytes.

Here, we performed the largest eQTL mapping to date in hepatocytes derived exclusively from AAs (n = 60) using our local ancestry adjusted method (LAMatrix) [28]. We identified AA-specific eQTLs (found only in our AA dataset) and overlapping eQTLs (found in both datasets) by comparing our results to those from the GTEx liver dataset. While the sample size for our study remains smaller than many GWAS efforts to date, human hepatocytes require human liver samples which are difficult to obtain, with AA donor livers being rarer still. We then characterized these AA-specific and overlapping eQTLs with respect to effect sizes, allele frequency in 1000 Genomes ancestral populations and enrichment in functional annotations. We implicated candidate genes for GWAS traits with known population divergences through AA-specific eQTLs and provided greater resolution to previous GWAS findings in the *SORT1* gene. These data provide a valuable resource to study genetic regulation of gene expression in AAs and highlight the utility of eQTLs to extend genomic studies of complex diseases in a minority population.

## Results

### Incorporation of local ancestry into eQTL mapping

Because AAs are an admixed population, the ancestral makeup of their genomes can differ by locus. Therefore, we applied our recently published method of adjusting for local ancestry (LA) in eQTL mapping (LAMatrix) [28]. This method has demonstrated increased power and control of false positives when gene expression differs by local ancestry [28]. We tested the genotype-gene expression associations within a *cis* region (1 Mb on either side of the gene), adjusting for sex, platform, batch, local ancestry and 10 PEER variables estimated from normalized expression values from 60 primary hepatocytes of AA ancestry (**Fig 1A**). The number of PEER variables was chosen to maximize the number of discovered eGenes (genes with at least one eQTL) as previously described (**S1 Fig**). Our eQTL analysis has identified 277 eGenes and 19,770 significant SNP-gene pairs (AA hepatocyte eQTLs, **Fig 1B, S3 Fig**) at a false discovery rate (FDR) threshold of 5% using hierarchical multiple-testing correction. Summary statistics for this analysis are publicly available through Figshare (https://figshare.com/projects/AA_hepatocyte_eQTLs/72635). Using conditional analysis, we identified 240 secondary eQTLs for 2 genes (*HCG4P7* and *PPIL3*, **S7E Fig, S7F Fig**). Of the 240 secondary eQTLs identified, 217 were not identified in our primary analysis.

We found 137 genes whose expression was associated with local ancestry at FDR < 0.1 and 65 genes at FDR < 0.05, providing support for the use of local ancestry adjustment in eQTL mapping. One such gene, Glutathione S Transferase 2 (*GSTA2)* (FDR = 0.03, **Fig 2A**), plays a pivotal role in drug metabolism and a potential role in acute anthracycline-induced cardiotoxicity [29]. Notably AAs have an increased risk of this drug-induced cardiotoxicity [30].

Using LAMatrix (FDR < 0.05), we were able to identify 1,179 additional eQTLs (**S12 Fig**) as compared to PC-adjusted eQTL mapping. By comparing the eQTLs unique to LA adjustment and eQTLs unique to PC adjustment, we found eQTLs unique to LA adjustment showed greater enrichment in the histone active markers such as H3K27ac (p = 4.52e-06, Fisher exact test) and H3K4me1 (p = 5.57e-08, Fisher exact test) in Roadmap liver tissue (**Fig 2B**), providing higher confidence of their regulatory effects.

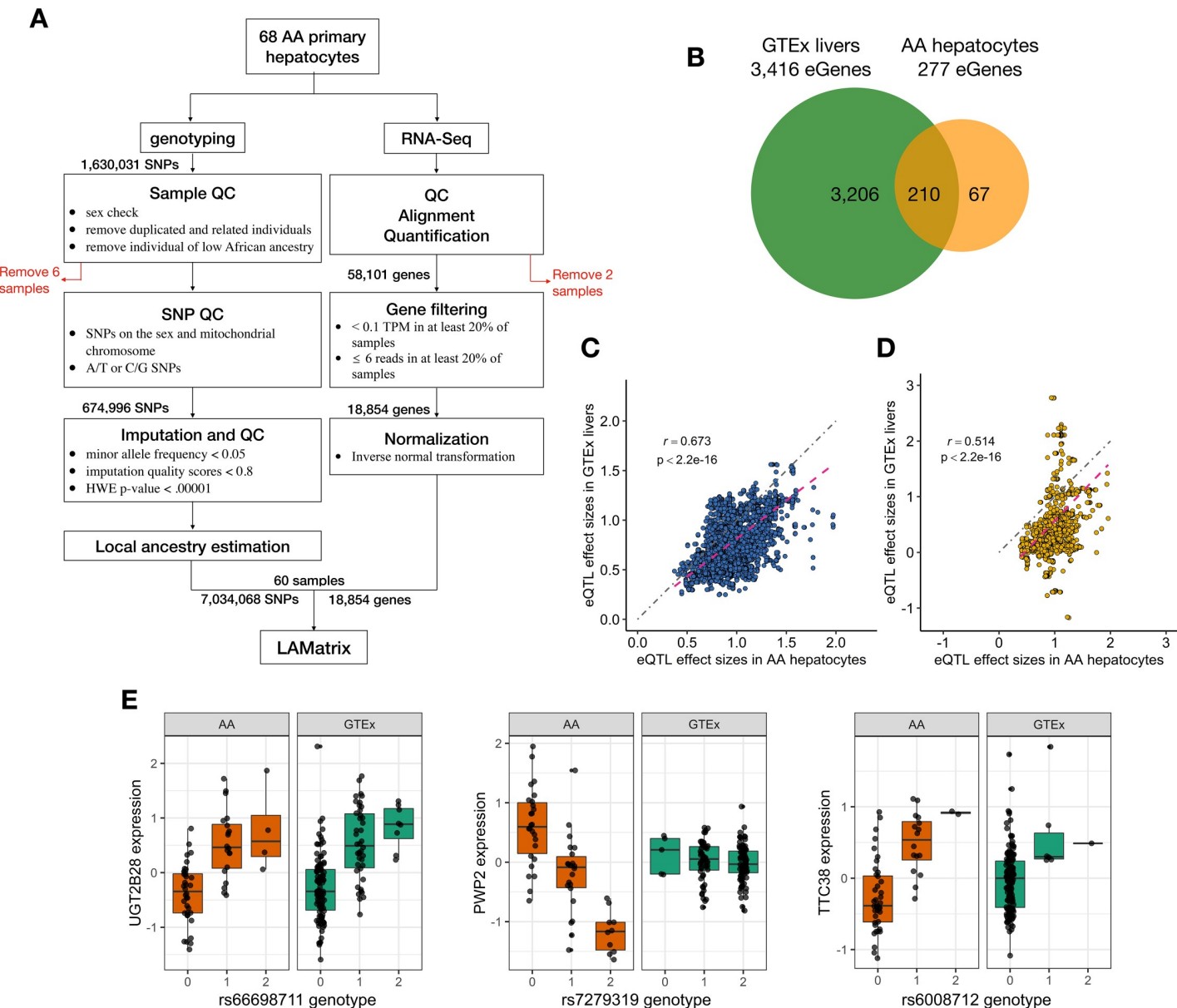

**Fig 1. Comparison of eQTLs mapped in AA hepatocytes (n = 60) with eQTLs mapped in GTEx livers (n = 153, v7).** To determine if the eQTLs discovered in AA hepatocyte were unique to our dataset or previously identified by the GTEx consortium, we compared our findings with this publicly available dataset. All significant eQTLs presented met the following criteria: FDR<0.05 and MAF>0.05 in each dataset. (A) Flowchart outlining the study design and analysis. (B) Venn diagram of eGenes discovered in AA hepatocytes as compared to eGenes in GTEx liver. (C) Comparison of absolute effect sizes of eQTLs found in both GTEx livers and AA hepatocytes (Overlapping eQTLs, Spearman correlation = 0.673, p-value<2.2e-16). Dotted magenta line represents the fitted regression line and dotted grey line represents the diagonal line. (D) Comparison of absolute effect sizes of eQTLs found only in AA hepatocyte dataset (AA-specific eQTLs), showing a lower correlation as compared with overlapping eQTLs (Spearman correlation = 0.514, p-value<2.2e-16). Dotted magenta line represents the fitted regression line and dotted grey line represents the diagonal line. (E) Examples of eQTLs discovered in AA hepatocytes. In the first panel, SNP, rs66698711, was significantly associated with the expression of *UGT2B28* in both GTEx and AA hepatocyte datasets (Overlapping eQTL). In the second panel, rs7279319 showed a negative effect on *PWP2* expression in AA hepatocytes but no effect in GTEx livers (AA-specific eQTL). The third panel, rs6008712, (an AA-specific eQTL) showed a positive correlation to *TTC38* expression in both cohorts but lacking statistical significance in GTEx due to the small MAF in the GTEx liver cohort.

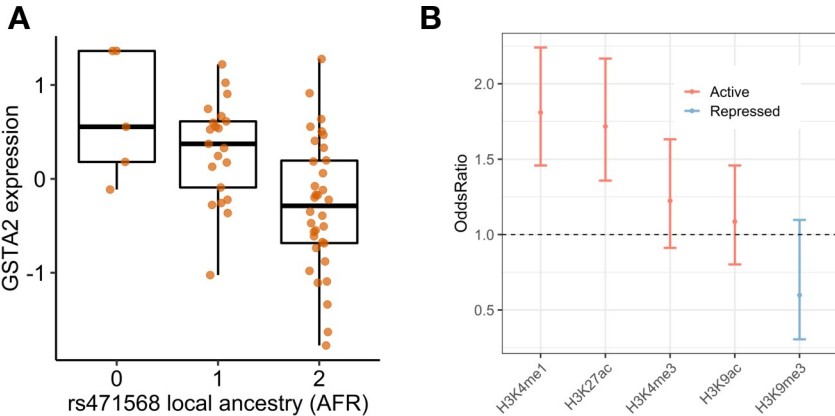

**Fig 2. eQTL mapping with local ancestry.** (A) The increased number of African ancestry alleles at rs471568 is associated with the decrease of *GSTA2* expression. (B) Enrichment of eQTLs unique to LA adjustment compared with eQTLs unique to PC adjustment in Roadmap histone modifications in liver tissue, showing the significant enrichment in active histone markers (e.g. H3K27ac and H3K4me3) and depletion in repressed marker (e.g. H3K9me3). These findings suggest that the LA adjustment when mapping eQTLs in admixed population may identify potential regulatory SNPs with more functional effects.

## Discovery of unique eGenes and eQTLs in AA hepatocytes and comparison to GTEx liver eQTLs

To identify eQTLs that may exert population-specific regulation on gene expression, we compared AA hepatocyte eQTLs with those found in the GTEx liver dataset (v7), filtered by MAF > 0.05 in GTEx livers. Of note, most of the liver samples within GTEx (v7) are of European ancestry, with only 15 livers obtained from individuals of African ancestry out of 153 samples in total (**S5 Fig**). Of the 277 eGenes discovered in the AA hepatocyte dataset, 210 were eGenes in the GTEx liver dataset, 8 were not tested for association to gene expression in GTEx and 59 were tested for association in GTEx but were not identified as eGenes (**Fig 1B, S7A, S7C and S7D Fig**). Of the 210 overlapping eGenes, only 25 share the same lead eQTL (the most significant eQTL for each gene), suggesting the genetic architecture of gene expression in European samples may differ from that in AA samples. The statistical significance of shared lead eQTLs is higher than distinct lead eQTLs (Wilcox rank-sum test, p = 0.001) in AA hepatocytes. We found a similar overlap when comparing AA hepatocyte eQTLs to eQTLs discovered with only the GTEx European liver samples (**S4 Fig**).

We identified eGenes unique to the AA hepatocyte eQTL analysis that are important for metabolism and blood coagulation. For example, we identified hepatocyte eQTLs for *F5* (encoding the protein Factor V) (**S7A Fig**), an important coagulation factor produced by the liver. Factor V Leiden is a well-known and clinically used genotype which is predictive of thrombotic risk. However, Factor V Leiden is at a very low allele frequency in African Americans [31], though African Americans carry a higher risk for thrombotic disease [32]. These regulatory variants were not discovered in GTEx liver eQTL analysis and may contribute to the increased risk of thrombotic disease in AAs.

Approximately 62.50% (12,355) AA hepatocyte eQTLs were significant eQTLs in the GTEx liver cohort (overlapping eQTLs) and 37.51% (7,415) eQTLs were unique to the AA hepatocyte cohort (AA-specific eQTLs, **S3 Fig**). When comparing the effect sizes of these overlapping eQTLs with effect sizes reported in GTEx, we found a strong correlation (**Fig 1C**, Spearman correlation = 0.673, p-value < 2.2e-16), confirming that there is shared genetic regulation between populations. However, correlation of effect sizes between AA-specific eQTLs with

those of GTEx liver were lower (**Fig 1D,** Spearman correlation = 0.514, p-value < 2.2e-16). Interestingly, the effect sizes of overlapping eQTLs were higher in AA hepatocytes compared to GTEx livers (p-value < 0.008).

An example of an overlapping eQTL, rs66698711, was significantly associated with the expression of *UGT2B28*, which is essential for the conjugation and elimination of toxic drug metabolites, in both GTEx and AA hepatocyte datasets (**Fig 1E**). In contrast, the association of rs7279319 to *PWP2* was only seen in AAs hepatocytes but not in GTEx livers, suggesting population-specific differences in the genetic regulation of *PWP2* (**Fig 1E**). Another population-specific eQTL example, rs6008712 was associated with *TTC38* expression in AAs and showed a trend towards association in GTEx livers (**Fig 1E**). Notably, this SNP is unique to African populations (**S13 Fig**) and has very low allele frequency (MAF = 0.026) in the GTEx liver cohort. Due to the small number of AA samples in the GTEx liver dataset, this eQTL was not detected as significant in GTEx liver.

We performed fine-mapping of overlapping eQTLs with CAVIAR [33] for each overlapping eGene (N = 210) and we estimated the casual set of SNPs underlying the eQTLs found in the AA hepatocytes analysis and the GTEx liver analysis. To remove the effect of sample size differences between cohorts, we randomly sampled 60 samples from GTEx European cohort to match the size of the AAs. AA hepatocyte eQTLs had significantly fewer number of eQTLs in the 95% causal set as compared to the GTEx liver subsets (median AA: 15, median GTEx: 173.5, Wilcox-rank sum test: p-value < 2.2e-16).

## Properties and functional characterization of AA hepatocyte eQTLs

We next sought to characterize the properties of overlapping and AA-specific eQTLs. AA-specific eQTLs have higher allele frequencies in 1000 Genome AFR populations than in the EUR populations (**Fig 3A**) (paired Mann-Whitney U test, p < 2.2e-16). The MAF distribution of AA-specific eQTLs in GTEx dataset is enriched for small values (**S6 Fig**). We also used fixation index (Fst) to directly measure the allele frequency differentiation between populations. AA-specific eQTLs have larger Fst compared with overlapping eQTLs (median Fst, AA-specific: 0.125; overlapping eQTLs: 0.117; Mann-Whitney U test, p = 1.82e-12, one-side, **S8 Fig**). This demonstrates the contribution of allele frequency differences to population-specific eQTL discoveries. It should be noted that we did not test for selective pressure in this analysis and that Fst was only used to characterize the difference in allele frequency between populations for AA-specific versus overlapping eQTLs.

To examine the functional roles of AA hepatocyte eQTLs, we tested the enrichment of eQTLs in histone modifications and transcription factors (TF) from the Roadmap Epigenomics and ENCODE projects (**Fig 2B and 2C**), respectively. We compared the number of eQTLs that overlap functional annotations to 1000 null sets of randomly sampled non-eQTL SNPs matched for MAF, LD score and distance to the TSS of the nearest genes. We showed that eQTLs were significantly enriched for active histone markers (H3K27ac, Bonferroni adjusted p = 5.45e-26 and H3K4me3, Bonferroni adjusted p = 2.44e-39) [34] and significantly depleted for repressed histone markers (H3K27me3, Bonferroni adjusted p = 3.21e-9) in Roadmap liver tissue (**Fig 2B**). However, The H3K9me3 was no longer significantly depleted in Roadmap HepG2 cell line (**S9B Fig**), indicating the potential difference in histone modifications between liver tissue and the HepG2 cell line. The enrichment stratified by eQTL group (overlapping and AA-specific eQTLs) showed a similar pattern to all AA hepatocyte eQTLs (**S9A Fig**). We also tested the enrichment of TF binding peaks using the ENCODE liver ChIP-seq data (**Fig 2C**). The top enriched TFs in ENCODE include HNF4A, which plays an important role in regulating metabolism, cell differentiation and proliferation in the liver [35].

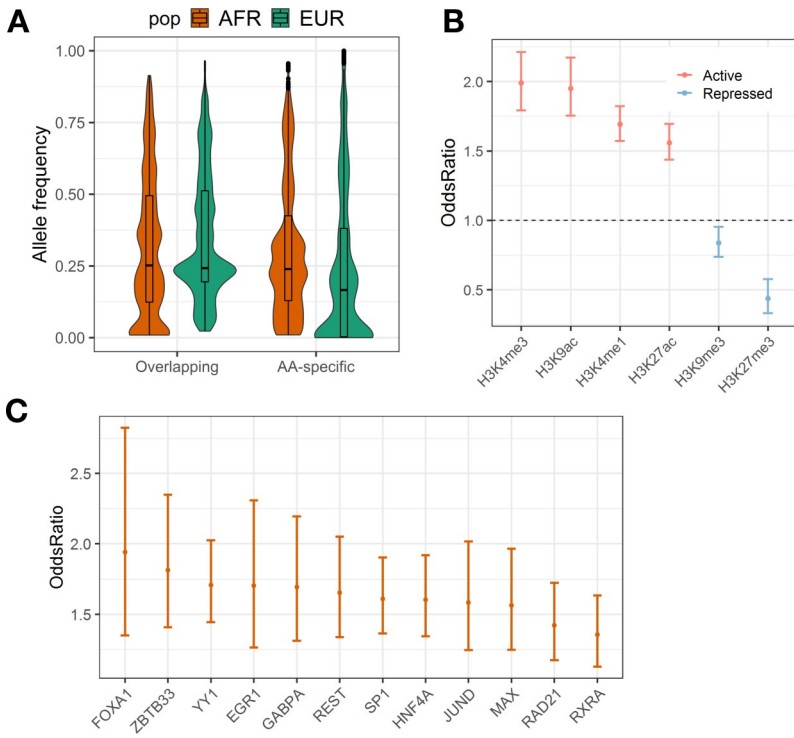

**Fig 3. Properties of overlapping and AA-specific hepatocyte eQTLs.** To characterize the properties of eQTLs discovered in our AA hepatocyte dataset, we use several public resources to explore different aspects of these potentially regulatory SNPs. (A) Violin plot of allele frequencies obtained from the 1000 Genome AFR and EUR populations for overlapping versus AA-specific eQTLs, which shows higher allele frequency for AA-specific eQTLs in AFR cohort as compared to EUR cohort (paired Mann-Whitney U test, p<2.2e-16). (B) AA hepatocyte eQTLs are significantly enriched in Roadmap liver annotations of active histone markers (H3K27ac and H3K4me3) and are significantly depleted in repressed histone markers (H3K9me3 and H3K27me3) (C) Enrichment of eQTLs within TF binding sites from ENCODE liver ChIP-seq data (ENCODE3 released version) showed significant enrichment for 12 TFs (Bonferroni corrected p value<0.05, 12 out of 16 in total).

## AA Hepatocyte eQTLs provide insights into GWAS associations

In order to determine the intersection of trait-associated SNPs with eQTLs in our study, we compared the AA hepatocyte eQTLs with the SNPs in the NHGRI-EBI catalog of published GWAS. GWAS variants or their tagging variants ($r^2 > 0.8$, 1000 Genomes CEU populations) from the GWAS catalog were used to determine which trait-associated SNPs intersected with our AA hepatocyte eQTLs. We identified 721 GWAS associations that intersect with AA hepatocyte eQTLs, providing evidence that eQTL target genes may play a role in trait associations (**Fig 4A**). The GWAS were categorized by Experimental Factor Oncology (EFO) terms and Lipid or lipoprotein measurement (FDR-corrected p-value = 8.87e-23), was significantly enriched (**Fig 4B**). This suggests that using eQTLs mapped in the relevant tissue context is critical for interpreting the association between traits and GWAS variants.

The number of intersecting hepatocyte eQTLs was significantly fewer per loci than the number of tagging SNPs per loci for GWAS variants (Wilcoxon rank sum test: p-value < 2.2e-16, median overlap with eQTLs = 34.5; median of all GWAS tagging variants = 137.5, **Fig 4C**). Forty-one GWAS variants had only one potentially causal SNP at the significant loci after intersecting with AA hepatocyte eQTLs. This finding suggests that where AA hepatocyte eQTLs intersect with previous GWAS findings, the use of these putative regulatory variants may narrow the number of potential causal variants at that site. As an example, GWAS first

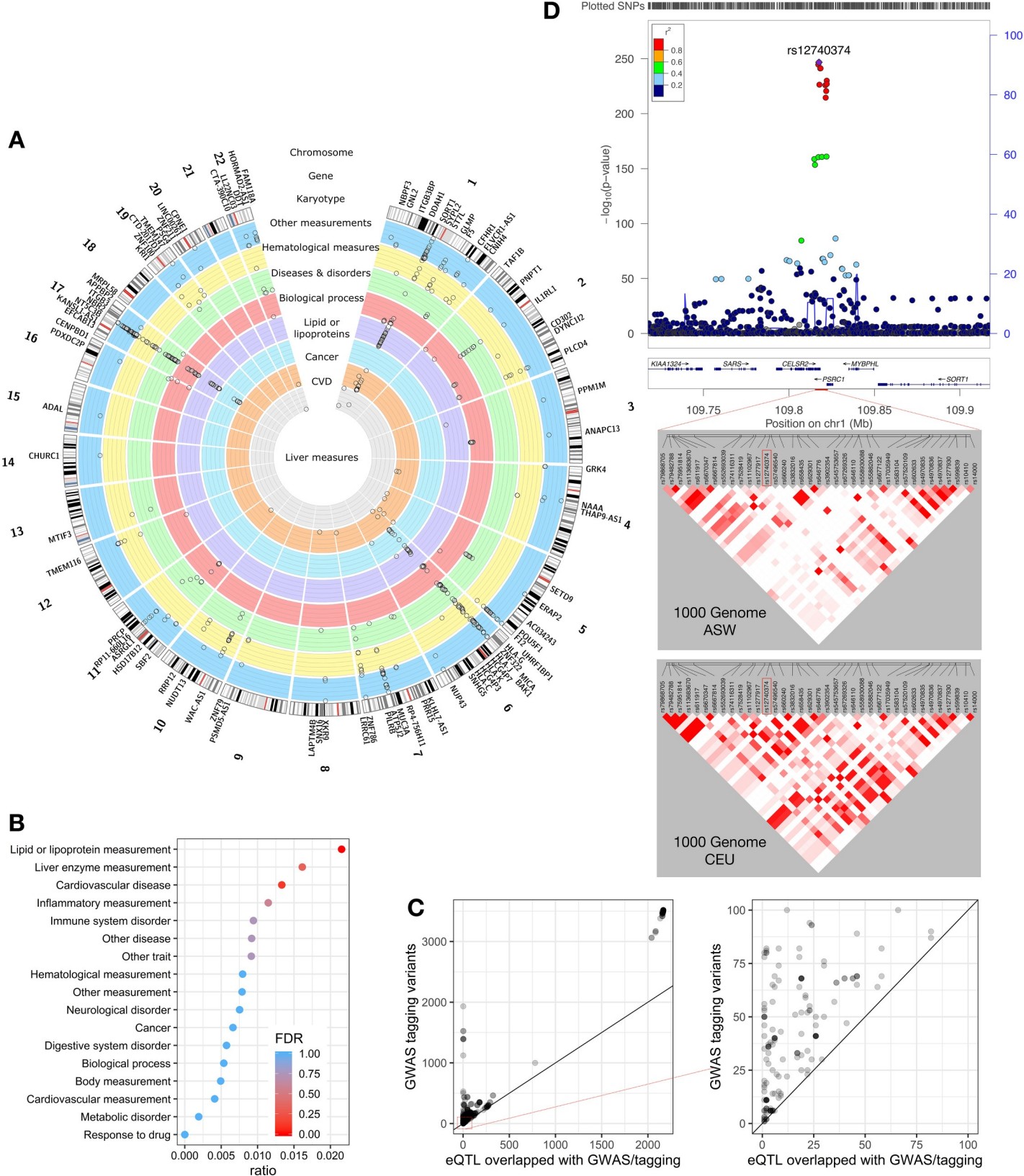

**Fig 4. Intersecting signals between significant SNPs (p-value < 5x10⁻⁸) in the NHGRI/EBI GWAS Catalog and AA hepatocyte eQTLs.** AA Hepatocyte eQTL were evaluated for intersection with NHGRI/EBI catalog GWAS SNPs. We identified 721 intersecting loci. (A) The Circos plot of hepatocyte eQTLs co-located with GWAS associations. Gene names external to the ring denote the 88 unique eGenes corresponding to eQTLs intersecting GWAS loci followed by the karyotype, reference hg38. The following inner rings show the association of 721 eQTLs discovered in hepatocytes for each EFO category (from outer to innermost ring): Other measurements (cerulean blue, n = 275); Hematological measurements, including inflammatory measurements (yellow, n = 93); Other diseases and disorders, including digestive system, immune system, metabolic and neurological disorders (green, n = 101); Biological processes, including body measurements (red, n = 70); Lipid or lipoprotein measurements (violet, n = 127); Cancer (sky blue, n = 19); Cardiovascular disease and measurements (CVD, orange, n = 32); Liver enzyme measurements (grey, n = 4). GWAS catalog v1.0.2 was used for analysis. All intersecting signals from the analysis are shown within each ring and depicted with open circles. The log p values for all 721 eQTLs identified from analysis were scaled from 0 to 1 and cut-off at 0.10 in the plot, with all eQTLs greater than 0.10 reaching the maxima. (B) GWAS variants intersecting with AA hepatocyte eQTLs are significantly enriched for "Lipid or lipoprotein measurement" EFO term as assessed by the hypergeometric test and FDR. Ratio (x axis) is the number of GWAS variants intersected with AA hepatocyte eQTLs over the total number of GWAS variants in each EFO category (y axis). (C) Figure comparing the number of tagging variants at each GWAS loci (y axis) and the number of tagging variants that intersect with AA hepatocyte eQTLs (x axis) at each GWAS loci. We used 1000 Genomes CEU population to extract tagging SNPs for all GWAS variants. Intersecting AA hepatocyte eQTLs have a lower number of tagging SNPs as compared to the same GWAS associated tagging variants at all loci. The right-hand panel shows the magnified region of 0–100 tagging SNPs, illustrating that several loci had only 1 eQTL while the GWAS associated SNP was tagged by many more SNPs. (D) AA hepatocyte eQTLs narrow the window of potential causal variants for a previously published GWAS of plasma LDL cholesterol level. The upper panel shows the locus zoom plot of LDL GWAS [36] labelling the top association (rs12740374, 1:109817590) in 1000 Genome EUR background (GRCh 37). The lower panel shows the LD blocks for SNPs around rs12740374 (1:109815074–109822509) calculated from 1000 Genome ASW and CEU population, showing a less extensive LD structure in the ASW genome. In the CEU genome there are 10 SNPs at LD ≥ 0.8 with the lead SNP (rs12740374), while there is only 1 SNP in high LD in the ASW genome.

mapped variants associated with lipid traits including low density lipoprotein cholesterol (LDL) levels to the *SORT1* gene, a gene encoding the sortilin protein, which is an intracellular sorting receptor. Subsequent functional validation provides evidence of the causal relationship between *SORT1* expression and LDL level [37]. Multiple variants in this region have been associated to lipid traits including rs660240, rs7528419, rs12740374 and rs602633 [36, 38], which all are eQTLs for the *SORT1* gene in the GTEx liver dataset. Notably, there are 34 liver eQTLs for *SORT1* in GTEx making it difficult to fine-map the causal variant for *SORT1* without functional assays. At the same locus there are only two AA-specific eQTLs (rs7528419 and rs12740374) for *SORT1* due to the to the less extensive LD block in Africans as compared with Europeans in this region (**Fig 4D**). One of the eQTLs, rs12740374, has been shown to sufficiently alter *SORT1* expression in luciferase assays [37]. To exclude sample size differences, we randomly sub-sampled the GTEx European liver dataset to match the sample size of AA hepatocyte data and performed eQTL mapping. The sub-sampled GTEx datasets averaged 19 eQTLs (S.D. = 3.2) for *SORT1* across 10 iterations. We performed fine-mapping with CAVIAR with the sub-sampled GTEx European and AA hepatocyte eQTL mapping results and found CAVIAR identified five eQTLs in the sub-sampled GTEx European cohort and only two eQTLs in the AA cohort in the 95% credible set. Additionally, after accounting for the increased number of SNPs within the *cis* window of *SORT1* in the GTEx dataset compared to the AA hepatocyte dataset (GTEx = 6,170, AA = 4,367), the use of AA hepatocyte eQTLs substantially decreased the number of potentially causal variants.

We next sought to investigate the contributions of overlapping versus AA-specific eQTLs to GWAS associations. We found 105 traits tagged by AA-specific eQTLs, 290 traits tagged by overlapping eQTLs, and 326 are tagged by both eQTL groups. The greater proportion of trait associations from overlapping eQTLs may be due to the biased representation of European cohorts in the GWAS catalog as well the growing number of GWAS with large sample sizes [20].

Using the 1000 Genomes YRI population to extract tagging variants results in fewer tagging variants (median of YRI: 45 and median of CEU: 137.5) but a similar enrichment pattern (Lipid or lipoprotein measurement, FDR-corrected p-value = 4.98e-07, Immune system disorder, FDR-corrected p-value: 1.75e-02, **S10 Fig**).

Our AA hepatocyte eQTLs identified candidate causal genes that were different from genes reported in the original GWAS. Platelet counts differ by population with AAs having higher platelet levels than Europeans [39]. High platelet count has been identified as a potential risk

factor for VTE in cancer patients [40, 41]. In contrast, thrombocytopenia is the most common hematologic abnormality in chronic liver disease [29]. Previously, a GWAS of platelet counts identified a risk variant, rs2251250, in chromosome 19 which was mapped to *ATP4A* [42]. Through our eQTL analysis, we found that rs2251250 was in LD with an overlapping eQTL, rs7599, for *TMEM147*, a transmembrane protein which regulates the M3 muscarinic acetylcholine receptor (M3R) encoded by *CHRM3* [43], which in turn, was implicated in VTE in a previous GWAS study [44]. Thus, *TMEM147* may serve as a potentially novel candidate gene for platelet count and VTE risk.

## Exon QTL mapping uncovers potential splicing variants

We further mapped exon QTLs to uncover the genetic regulation of alternative exon usage. We identified 1,284 exon segments with significant genetic regulation (eExons) and in total 75,002 exon QTLs. Among the 730 genes for all eExons, 223 genes were also identified as eGenes in eQTL mapping and 507 genes were unique to the exon QTL analysis. A majority, 51.2% (38,390 out of 75,002), of the exon QTLs were also identified as eQTLs.

As an example, a previously identified and clinically significant exon QTL is rs776746 for *CYP3A5*. This SNP creates a cryptic splice site in intron 3 resulting in a premature termination codon and a nonfunctional CYP3A5 protein (**Fig 5A**) [45]. This SNP also differs in allele frequencies between global populations (G: 18% in 1000 Genome AFR and 94% in EUR population). As a second example, rs1019299 is an exon QTL for *STRADB*, which encodes a protein in serine/threonine protein kinase *STE20* subfamily. This exon QTL is associated with the expression of exon segment 5 (**Fig 5B**) and is a synonymous coding variant. Notably *CYP3A5* and *STRADB* are not eGenes in our eQTL mapping analysis. These identified eExons suggest exon QTL mapping complements eQTL mapping analysis and provides greater resolution to reveal alternative splicing events.

## Discussion

Regulatory variants for molecular and phenotypic traits have demonstrated great heterogeneity across human populations. However, a lack of diversity persists in current genetic studies. For example, non-Europeans correspond to only 19% of individuals involved in studies found in the GWAS catalog [20] and less than 5% of subjects are non-European and non-Asian [46]. The racial disparity in eQTL studies is even more striking, especially in tissues relevant to disease and pharmacogenomic traits. Here, we performed the first eQTL mapping in a disease-relevant tissue, liver, and in an underrepresented population and discovered novel eQTLs for disease-related genes. While other eQTL studies have AA samples in their analysis [47], the number of samples has been much smaller and gene expression was quantified via gene expression microarrays. Many of these novel eQTLs were discovered because of the higher allele frequency of these variants in African ancestry populations, which increased our power to detect them even in our smaller dataset. Consequently, we were able to provide functional evidence and implicate novel candidate genes for GWAS associations as well as show a decrease in the number of potentially causal variants. This demonstrates the advantages of mapping eQTLs in individuals of African descent and other diverse populations, in which greater genetic diversity, differences in allele frequency and LD structure are likely to aid in fine-mapping novel regulatory variants. With the shorter span of LD found in populations of African ancestry, it is possible to further refine the location of causal regulatory variants found thorough GWAS. For example, the peak of association for *PRSS45* is shorter in our AA hepatocytes eQTL results than in GTEx liver eQTL results (**S7B Fig**). Regardless, these findings should be interpreted with caution given eQTL mapping tests for correlation, as opposed to causation.

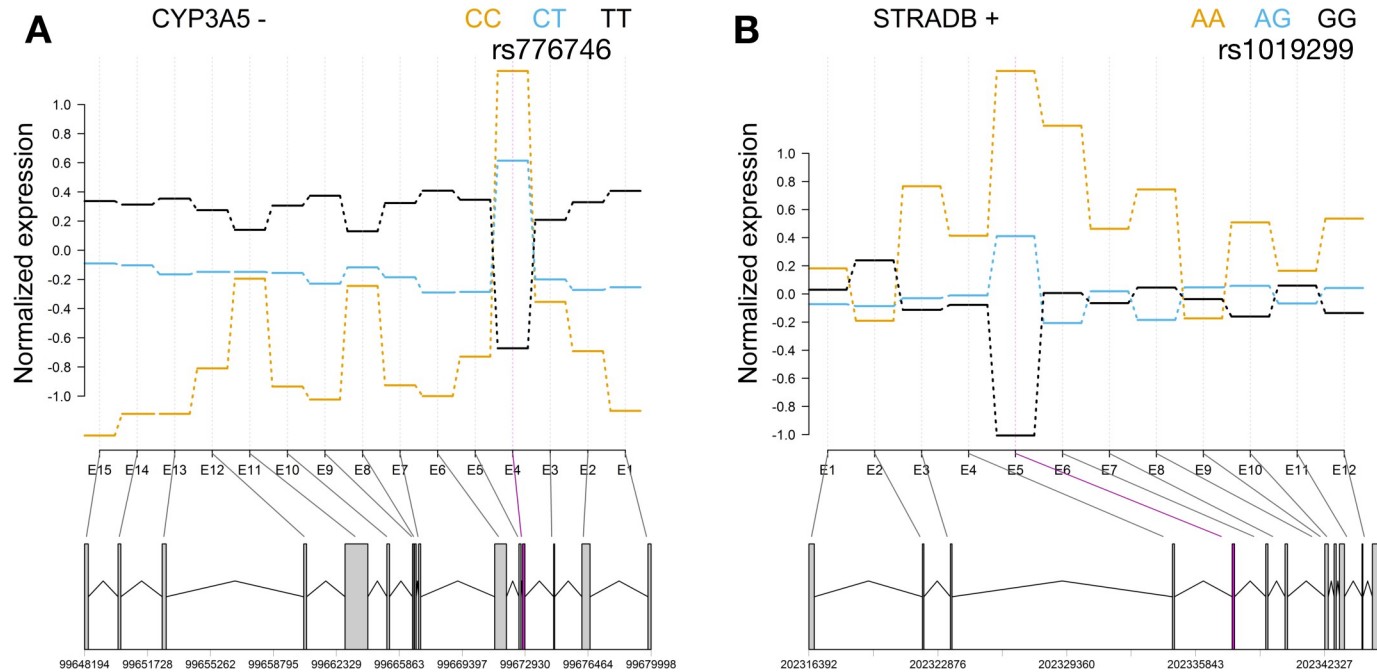

**Fig 5. Exon QTLs provide insights into splice events in *CYP3A5* and *STRADB*.** This figure shows the normalized exon expression by genotype for the exon segment tracts below. The magenta color denotes the eExon and the grey color denotes non-eExons. The plus and minus symbol next to the gene represents the strand. (A) An exon QTL, rs776746, creates a cryptic splice site in intron 3 resulting in a premature termination codon and a nonfunctional CYP3A5 protein. (B) The *STRADB* gene encodes a protein in serine/threonine protein kinase STE20 subfamily and does not have a significant eQTL from eQTL mapping analysis. One exon QTL, rs1019299, is associated with the expression level of exon 5 and is a synonymous variant for *STRADB* gene.

We showed that most of the AA hepatocyte eQTLs were also found in the GTEx liver dataset with a significant correlation of effect sizes between these two datasets. This is in concordance with a previous study, where extensive eQTL sharing was found across Africans, Asians and Europeans, even though the correlation of effect sizes is lower for AA-specific eQTLs [24]. Furthermore, AA-specific eQTLs have larger population differentiation as measured by Fst. Therefore, all three factors–LD structure, effect size heterogeneity and allele frequency differences–may underline population specific regulatory variants [28]. It would be of interest to further distinguish the effect of each component and elucidate its impact on causality by evaluating this data using massively parallel functional assays [48]. Moreover, our study suggests that accounting for all three factors is critical to increase transferability of gene expression prediction models across populations. Accounting for interactions between genotype and local ancestry as well as constructing predictive models of gene expression with shared variants may be useful.

In the enrichment of GWAS ontologies among AA hepatocyte eQTLs, the category "Response to Drug" was not enriched although liver is the key organ for drug metabolism. It is possible that the eQTLs related to drug response are discoverable only under specific contexts as seen previously for immune response eQTLs [11]. Additionally, drug response GWAS (n = 541) represent 0.63% of the overall GWAS catalog with many of the significant findings representing coding variants, which may not change gene expression but change the functional activity of the protein [49].

Our study has several limitations. Firstly, the sample size of our AA cohort is relatively small, and thus limited the power to discover AA-specific eQTLs with small to moderate effect sizes. For example, *VKORC1* encodes a major subunit of vitamin K epoxide reductase complex

and is a target enzyme for warfarin, while rs9923231 is known to regulate *VKORC1* activity and warfarin sensitivity [50]. Although this SNP is a significant eQTL in GTEx, its association in the AA hepatocyte data does not rise above FDR threshold (FDR-corrected p-value = 0.23). Secondly, the comparison between AA hepatocytes to GTEx liver is not completely straightforward. The expression quantification in GTEx was performed in post-mortem liver tissues which are made up of various cell types including Kupffer cells and endothelial cells, rather than only hepatocytes. The contribution of these additional cell types may impact eQTL mapping in this heterogeneous tissue. Similarly, we used post-mortem samples, but we extracted living hepatocytes, thus limiting our analysis to one cell type. Others have noted an effect of post-mortem time on gene expression [51–53]. These studies were conducted in flash frozen tissues and the effect of post-mortem time on primary hepatocytes is unknown. Therefore, eQTLs found in our AA cohort may represent either AA-specific eQTLs or hepatocyte-specific eQTLs. As noted in the results, only 8 eGenes were found in the AA hepatocyte dataset, which were not expressed in the GTEx liver dataset. On the other hand, our data may serve as a valuable resource to investigate hepatocyte-specific eQTLs in contrast to multicellular expression represented in liver eQTLs.

In conclusion, we identified a substantial number of novel regulators of gene expression in AA hepatocytes and highlight the utilities of eQTL mapping in an underrepresented population. Leveraging distinct LD structures with respect to molecular phenotypes across diverse populations will improve our ability to assess mechanisms of transcriptional regulation and genetic contributions to diseases.

## Methods

### Ethics statement

This study was deemed non-human research by the Northwestern Institutional Review Board.

### Cohort

A total of 68 AA primary hepatocyte cultures were used for this study. Cells were either purchased from commercial companies (BioIVT, TRL, Life technologies, Corning and Xenotech), or isolated from cadaveric livers using a modified two-step collagenase perfusion procedure. Liver specimens were obtained through collaborations with Gift of Hope, which supplies non-transplantable organs to researchers. We excluded any livers with active cancer or with a history of hepatocarcinoma. Primary hepatocytes were isolated and cultured as previously described in Park et al [54]. The workflow of genotyping QC and RNA sequencing QC is in **Fig 1A**.

### Genotyping and imputation

DNA was extracted from each hepatocyte line using Gentra Puregene Blood kit (Qiagen). All DNA samples were bar coded. SNPs were genotyped using Illumina Multi-Ethnic Genotyping array (MEGA) at the University of Chicago Functional Genomics Core using standard protocols. Genotyping outputs were created by Genome Studio using 0.15 GenCall score cutoff.

Sex check by PLINK [55] was performed to identify individuals with discordant sex information. Duplicated or related individuals were identified using identity-by-descent (IBD) method with a cutoff score of 0.125 indicating third-degree relatedness. Additionally, subjects that did not cluster with the AA samples on the PCA plot were removed. A total of 6 individuals were excluded after sample and genotyping QC analysis, leaving 62 individuals.

The following SNPs were excluded: SNPs on the sex and mitochondrial chromosome, A/T or C/G SNPs which may introduce flip-strand issues, SNPs with missing rate > 5% or failed Hardy-Weinberg equilibrium (HWE) tests (p < .00001), leaving 674,996 SNPs. Genotypes were phased using SHAPEIT [56] and imputed with IMPUTE2 [57] using all reference populations from 1000 Genome phase 3. After imputations, SNPs were excluded for minor allele frequency < 0.05, imputation quality scores < 0.8, and HWE p-value < .00001, leaving 7,180,502 SNPs in the analysis. Imputed genotypes were converted to the gene dosages.

## RNA-sequencing and quality control

Total RNA was extracted from each primary cell culture after three days of plating using the Qiagen RNeasy Plus mini-kit. Samples with RNA integrity number (RIN) score > 8 were considered for sequencing. Libraries were prepared for sequencing using the TruSeq RNA Sample Prep Kit, Set A (Illumina catalog # FC-122-1001) according to manufacturer's instructions. The cDNA libraries were prepared and sequenced using both Illumina HiSeq 2500 and HiSeq 4000 machines by the University of Chicago's Functional Genomics Core to produce single-end 50 bp reads with approximately 50 million reads per sample.

Quality of the raw reads from FASTQ files was assessed by FastQC (v0.11.2) [58]. The fastq files with per base sequence quality threshold of > 20 across all bases were used. Reads were aligned to human Genome sequence GRCh38 and Comprehensive gene annotation (GEN-CODE version 25) using STAR 2.5 [59]. Only uniquely mapped reads were retained and indexed by SAMTools 1.2 [60]. Nucleotide composition bias, GC content distribution and Coverage skewness of the mapped reads were further assessed by read_NVC.py, read_GC.py and geneBody_coverage.py from RNA-SeQC (2.6.4), respectively. Samples without nucleotide composition bias or coverage skewness and with normally distributed GC content were reserved. Lastly, Picard CollectRnaSeqMetrics was applied to evaluate the distribution of bases within transcripts. Fractions of nucleotides within specific genomic regions were measured. Samples with > 80% of bases aligned to exons and UTRs regions were retained for analysis.

## Gene expression quantification

We used a collapsed gene model following the GTEx isoform collapsing procedure [61]. Reads were mapped to genes referenced with Comprehensive gene annotation (GENCODE version 25) to evaluate gene-level expression using RNA-SeQC [62]. HTSeq [63] raw counts were supplied for gene expression analysis using Bioconductor package DESeq2 (version1.20.0) [64]. Counts were normalized using regularized log transformation and principal component analysis (PCA) was performed in DESeq2. PC1 and PC2 were plotted to visualize sample expression pattern. Two samples with distinct expression patterns were excluded as outliers.

We normalized the gene expression by trimmed mean of M-values normalization method (TMM) implemented in edgeR [65]. We calculated the TPM (transcript per million) by first normalizing the counts by gene length and then normalizing by read depth [66]. Gene expression values were filtered based on expression thresholds < 0.1 TPM in at least 20% of samples and ≤ 6 reads in at least 20% of samples. The expression values for each gene were normalized across samples with inverse normal transformation. We remapped the gene coordinates to hg19/GRCh 37 (GENCODE version 19).

## eQTL Mapping using Matrix eQTL

We performed eQTL mapping for 60 samples that passed both RNA sequencing and genotyping QCs. To correct for both measured and hidden confounders to gene expression analysis, we calculated Probabilistic Estimation of Expression Residuals (PEER) factors [67] for

normalized gene expression values and exon expression values using PEER R package. We reported eQTL mapping results using the top 10 PEER factors, sex, sequencing platform, batch, and the first PC calculated from pre-imputation LD-pruned genotypes as covariates. The correlation structure of covariates is shown in **S2 Fig**. The linear regression coefficient for each SNP (within a 1 mb of the gene transcriptional start site and 1 mb of the gene end site) with normalized gene expression was estimated with Matrix eQTL [68].

We used the hierarchical correction method to call significant eQTLs as previously described [69, 70]. Firstly, p values of all cis SNPs are adjusted for multiple testing for each gene using Benjamini and Yekutieli (BY) method [71] as the locally adjusted p values. Secondly, the minimum BY-adjusted p values for all genes are corrected using Benjamini and Hochberg method (BH) [72] as the globally adjusted p values (BY-BH p values). Lastly, for a chosen threshold (here we use, 0.05), we found the largest BY-BH p values under that threshold and the corresponding BY-adjusted p value. This BY-adjusted p-value is used as the threshold to call significant eQTLs.

Enriched pathway analysis was performed with gProfiler [73] using all the expressed genes in AA hepatocytes as the background for queried eGene list.

## Mapping eQTL with local ancestry information

We estimated the local ancestry of AA hepatocyte genotypes with RFMix [74] with YRI and CEU samples from 1000 Genome phase 3 as the reference populations for 7,034,068 SNPs, using a window size of 0.2 Mb. The average local ancestry across the genome correlated with the first PC (**S11 Fig**). We applied the method, LAMatrix, which performs linear association between gene expression and genotype while adjusting for both individual-specific covariates (e.g. sex) and locus-specific covariates (e.g. local ancestry etc.). We mapped eQTLs with the LAMatrix approach using the same covariates used in the previously described eQTL mapping with Matrix eQTL excluding the first PC and applied hierarchical multi-testing correction to identify significant eQTLs (FDR<0.05). We maximized the number of discovered eGenes by testing a range of numbers of PEER variables (**S1 Fig**).

We also tested the effect size of local ancestry with gene expression adjusting for top 10 PEER factors, sex, sequencing platform, and batch for each SNP. We extracted the most significantly associated LA block within the cis region of the gene and applied FDR correction to identify gene expression signatures associated with local ancestry.

## Conditional analysis

We identified the most significant eQTL (lead eQTL) for each eGene and regress the effect of the lead eQTL to obtain the residual gene expression as Jansen et. al. [75]. We performed eQTL mapping with the residual expression and SNPs within 1Mb of the gene. We adjusted p-values of all *cis* SNPs for each gene using Benjamini and Yekutieli (BY) method and used the same BY threshold to call secondary eQTL as was done with primary eQTLs.

## Comparison between AA hepatocyte eQTLs and GTEx Liver eQTLs

All tested gene-SNP pairs and significant eQTL pairs in the GTEx liver cohort (version 7) were downloaded from the GTEx portal. The GTEx dataset includes 153 liver samples with available genotype and gene expression data of which only 15 of African descent (based on phenotype file, verified by PCA, **S5 Fig**). The GTEx dataset was filtered by SNPs with minor allele frequency (MAF)>0.05, leaving 6,453,712 SNPs. A total of 311,967 eQTLs and 3,416 eGenes in the GTEx liver were retained with this filtering.

We overlapped the AA hepatocyte eQTLs with the GTEx eQTLs to identify overlapping eQTLs and AA-specific eQTLs. AA hepatocyte eQTLs that were also significant in the GTEx dataset and have concordant direction of effect were identified as overlapping eQTLs. AA hepatocyte eQTLs that were not found in GTEx were identified as AA-specific eQTLs.

To compare AA hepatocyte eQTLs with GTEx liver eQTLs in the European background, we selected 127 samples that are of European ancestry from GTEx liver samples (**S5 Fig**) and performed eQTL mapping with FastQTL [76] as described in GTEx v7. We performed 1000 permutation to identify the threshold of eGene discovery and selected corresponding eQTLs at FDR<0.05. We found a similar pattern of overlapping results as the results using all samples. (**Fig 1B** and **S3 Fig**). We also randomly sampled 60 samples from GTEx European liver samples to match the sample size of AA hepatocytes and performed eQTL mapping as described above (total of 10 iterations). We identified the number of overlapping and unique eQTLs/ eGenes averaged across 10 iterations.

We used CAVIAR (CAusal Variants Identification in Associated Regions) [33] to statistically fine-map eQTLs in AA hepatocytes and GTEx subsampled eQTLs separately. We applied CAVIAR to the *cis* eQTL mapping z scores and LD for each overlapping eGene while setting the number of causal variants to 1. CAVIAR estimates the set of SNPs that account for 95% of the posterior probability of the causal variants at each locus.

## Properties and functional characterization of eQTLs

Allele frequencies of eQTLs in AFR and EUR populations were acquired from 1000 Genome phase 3. We calculated the Fst for eQTLs using 1000 Genomes YRI and CEU populations with GCTA software [77]. We tested differences in Fst between overlapping eQTLs and AA-specific eQTLs using the Wilcox rank test.

We calculated the LD score of each SNP with LD Score Regression (LDSC) [78] using the window size of 1 cM with AA hepatocyte genotype data. LD score is the correlation of the index SNP with all other neighboring SNPs, representing the locus-specific LD structure.

To test the enrichment of eQTLs in histone markers and transcription factor (TF) binding, we sampled 1000 null set of SNPs matching AA hepatocyte eQTLs by MAF, LD score and distance to transcriptional start site (TSS) of nearest gene within 10 quantile bins in AA hepatocyte genotypes. For the histone marker enrichment analysis, we used the ChIP-Seq and consolidated narrow peak results from Roadmap for liver tissue (E066) and HepG2 cell line (E108). For the TF binding enrichment, we used ENCODE3 released version ChIP-Seq and the optimal idr thresholded peaks (narrow peak) in liver for 16 TFs.

We averaged the number of null SNPs that are overlapped with each of these epigenetic annotations across 1000 random sets and tested the enrichment with Fisher exact test comparing the number of eQTLs overlapped with epigenetic annotations versus the average number of null SNPs overlapped with epigenetic annotations. We corrected the Fisher exact test p-value using Bonferroni method and reported the enrichment with adjusted p value<0.05.

## Overlap eQTLs with GWAS hits

To investigate how eQTLs can inform the molecular mechanisms underlying GWAS association findings, we downloaded NHGRI/EBI GWAS Catalog file (v.1.0.2, 2019-03-22) [79] and kept associations that passed the genome-wide significant level (p<5e-8). We remapped the rsids from Build38 to Build37 using Ensembl API. We used 1000 Genomes YRI and CEU population to extract all the variants in LD with the independent GWAS variants (r2>0.8). We categorized traits of corresponding GWAS into 17 groups representing larger, empirically determined and ontology-based trait categories [79]. We reported the overlapping signals as

the eQTLs that are the GWAS variants/tagging variants. This method was used instead of more commonly analytical methods, such as *coloc* [80], because of the known LD differences in our AA hepatocyte data as opposed to the GWAS catalog data which were performed in predominantly European subjects.

## Exon QTL mapping

To map SNPs that associated with exon-level expression, we flattened the gene annotations (GENCODE25) by merging intervals of the overlapping exons using the GTEx pipeline [61] and counted the exon reads with HTSeq [63] (n = 327,304). We then filtered the exons with the same filtering threshold as the gene-level data and converted the counts to RPKM measurements (n = 183,193). We quantile normalized the RPKM across samples and used inverse normal transformation to normalize each gene. We incorporated 10 PEER variables estimated from exon expression, sex, platform, batch and the top PC as covariates. We performed Lift-Over to convert the exon genome positions from hg38 to hg19.

We mapped the exon QTLs with FastQTL for a window size of 500 kb. We performed 1000 permutations to determine the threshold of eExon discovery and selected corresponding exon QTLs at FDR<0.05.

## Supporting information

**S1 Fig. Plot of discovered eGenes with respect to the number of PEER variables added to the eQTL mapping model.** We tested different numbers of PEER variables as covariates in LA adjusted eQTL mapping of AA hepatocytes in order to maximize the power to discover eGenes. Using the hierarchical multi-testing correction method and the threshold of FDR<0.05, we found that 10 PEERs gave the largest number of discovered eGenes and this result was used in the downstream eQTL mapping.
(TIF)

**S2 Fig. Pair-wise correlation of the covariates.** We plot the pair-wise correlation of the covariates used in the eQTL mapping model. The lower triangle shows the pair wise scatter plot with the red line representing the fitted regression line. The diagonal represents the histogram of each covariant. The upper triangle shows the Pearson correlation r for each pair of covariates with the significant correlation highlighted with red stars (p< 0.001 = ***, p<0.01 = **, p<0.05 = *, p<0.1 = dot). Due to the high correlation between PC1, PC2 and PC3 estimated from genotype data (also can be seen from **S5 Fig**), only PC1 was used as a covariate to account for population substructure.
(TIF)

**S3 Fig. Venn Diagram of the overlapping of eQTLs discovered in AA hepatocytes and GTEx livers.** Approximately 62.50% (12,355) AA hepatocyte eQTLs were significant eQTLs within GTEx liver dataset and 37.51% (7,415) eQTLs were unique to the AA hepatocyte dataset.
(TIF)

**S4 Fig. Venn diagram of the overlapping of eGenes and eQTLs discovered in AA hepatocytes and GTEx livers of European ancestry.** We selected 127 samples that are of European (EUR) ancestry in GTEx liver cohort (**S5 Fig**) and performed eQTL mapping with FastQTL. We did 1000 permutation to identify the threshold of eGene discovery and selected corresponding eQTLs at FDR<0.05. We found a similar pattern of overlapping as the main results present in the manuscript (**Fig 1B** and **S3 Fig**).
(TIF)

**S5 Fig. Principal Component Analysis of AA hepatocytes and GTEx liver genotypes merged with HapMap 3 global populations.** The GTEx liver samples are shown by the race/ethnicity provided in the GTEx phenotype file. The vertical line (PC1>0.016) was used to identify GTEx liver samples of European descent (n = 127). The GTEx AA samples (n = 15) and AA hepatocyte samples (n = 60) lie in the cluster between HapMap CEU and YRI. (TIF)

**S6 Fig. Histogram of MAF in GTEx and AA hepatocytes for AA-specific eQTLs.** Among 7,415 AA-specific eQTLs, 4,291 were available in GTEx dataset. The comparison of the MAF (calculated in each cohort) show that AA-specific eQTLs are skewed towards small MAF in GTEx. Hence, the GTEx liver dataset may not have been powered to detect these SNPs as eQTLs. (TIF)

**S7 Fig. Regional plot of AA hepatocyte eQTLs.** Regional plot for AA hepatocyte eGenes shows different patterns of associations. Plotted are the locusZoom plots showing the -log P values of cis SNPs to the labelled gene in either AA hepatocytes or GTEx livers. **A)** The unique eGene, *F5*, has associations with genetic variants specific to AA hepatocytes. **B)** In the *PRSS45* gene, the AA hepatocyte eQTLs are in shorter stretches of LD than GTEx liver eQTLs. **C)** eQTLs for *KIF5A* are only found in AA hepatocyte dataset but not in GTEx liver dataset. **D)** The GWAS variant, rs7903847, which is associated with granulocyte percentage of myeloid white cells, is in LD with an AA-specific eQTL, rs10786336, for *RRP12* gene. This SNP is not an eQTL in the GTEx liver analysis. **E)** and **F)** show the eGenes with secondary eQTLs. (TIF)

**S8 Fig. Violin Plot of Fst values for overlapping and AA-specific eQTLs.** We calculated Fst values using 1000 Genome phase 3 data using YRI and CEU populations with GCTA. The Fst values for overlapping and AA-specific eQTLs were compared. Here shows that the AA-specific eQTLs have higher Fst than the overlapping eQTLs (Mann-Whitney U test, p = 1.82e-12, one-side). (TIF)

**S9 Fig. Enrichment of AA hepatocyte eQTLs in Roadmap histone modifications and Encode TF binding by eQTL category.** We tested the enrichment of AA hepatocyte eQTLs, overlapping eQTLs, and AA-specific eQTLs in Roadmap histone modifications mapped in liver tissue (**A**) and HepG2 cell line (**B**) compared with a matched null SNPs set. The H3K9me3 is no longer significantly depleted in HepG2, suggesting the different landscapes of histone modifications between liver tissue and HepG2 cell line. (**C**) shows the enrichment in TF binding in hepatocytes in ENCODE for all, overlapping and AA-specific eQTLs. (TIF)

**S10 Fig. GWAS EFO enrichment using 1000 Genome YRI background.** We also used 1000 Genomes YRI population to prune GWAS variants and find GWAS tagging variants (related to **Fig 3B**). The corresponding GWAS were significantly enriched in the following ontologies: Lipid or lipoprotein measurement FDR-corrected p value = 4.98e-07, Immune system disorder FDR-corrected p value: 1.75e-02. (TIF)

**S11 Fig. Correlation of average LA ancestry across the genome and the first PC.** The LA ancestry was estimated with RFMix and averaged across the genome. Here we showed a high

correlation between the averaged LA ancestry and the first PC in AA hepatocyte data.
(TIF)

**S12 Fig. Comparison of eQTLs identified with LA adjustment and PC adjustment.** We mapped eQTLs using LAMatrix which adjusts for local ancestry instead of PC in the eQTL mapping for AAs. We identified 1,179 additional eQTLs representing potential novel eQTLs found by accounting for local ancestry structure.
(TIF)

**S13 Fig. Allele frequency of rs6008712 in 1000 Genomes Project Phase 3.**
(TIF)

**S14 Fig. Principal Component Analysis of expression counts showing two outlier samples.** Gene counts were normalized using regularized log transformation and principal component analysis (PCA) was performed with DESeq2. PC1 and PC2 were plotted to visualize sample clustering.
(TIF)

## Acknowledgments

The authors appreciate Yilin Xu and Eric Gamazon for the critical reading of the manuscript and Yinan Zheng for the helpful discussions.

## Author Contributions

**Conceptualization:** Yizhen Zhong, Minoli A. Perera.

**Data curation:** Tanima De, Cristina Alarcon, Bianca Lec.

**Formal analysis:** Yizhen Zhong.

**Funding acquisition:** Minoli A. Perera.

**Methodology:** Tanima De.

**Visualization:** Yizhen Zhong, C. Sehwan Park.

**Writing – original draft:** Yizhen Zhong, Minoli A. Perera.

**Writing – review & editing:** C. Sehwan Park, Minoli A. Perera.

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
