## [Decision Letter · Decision Letter 0]

4 Oct 2019

Dear Dr Perera,

Thank you very much for submitting your Research Article entitled 'Discovery of novel hepatocyte eQTLs in African Americans' to PLOS Genetics. Your manuscript was fully evaluated at the editorial level and by independent peer reviewers. The reviewers appreciated the attention to an important problem, but raised some substantial concerns about the current manuscript. Based on the reviews, we will not be able to accept this version of the manuscript, but we would be willing to review again a much-revised version. We cannot, of course, promise publication at that time.

If you decide to revise the manuscript for further consideration at PLOS Genetics, please aim to resubmit within the next 60 days, unless it will take extra time to address the concerns of the reviewers, in which case we would appreciate an expected resubmission date by email to plosgenetics@plos.org.

[LINK]

We are sorry that we cannot be more positive about your manuscript at this stage. Please do not hesitate to contact us if you have any concerns or questions.

Yours sincerely,

Christopher R Gignoux, Ph.D.

Guest Editor

PLOS Genetics

Scott Williams

Section Editor: Natural Variation

PLOS Genetics

Reviewer's Responses to Questions

**Comments to the Authors:**

Reviewer #1: The manuscript by Zhong and colleagues describes an effort to map eQTLs in hepatocyte cell lines derived from African Americans (AAs). The authors note that AAs are understudied in eQTL mapping efforts, and that eQTL mapping in AAs could potentially identify novel genetic architecture of gene expression. Since hepatocytes reside in the liver, an important organ for drug metabolism, the authors argue that mapping eQTLs in hepatocytes in particular may shed light on drug responses that differ by ethnicity. The authors use the MatrixEQTL software package to map eQTLs in 60 hepatocyte cell lines from AAs. They identify eQTLs (and eGenes and eExons) and then proceed to characterize the eQTLs using publicly available functional genomic information from the ENCODE project. They further characterize AA-specific eQTLs by way of comparison against GTEx, and they also characterize eQTLs by local ancestry using their LAMatrix software package. The authors present results that show eQTLs with clear dosage effects and sometimes dramatic differences in MAF between their AA cohort and GTEx.

Overall, I feel that the analyses in the paper are structured logically and that the analyses provide convincing evidence for the existence of population-specific eQTLs in AAs. Despite the sample size of n = 60, which some might argue is a bit small, the authors show fairly convincing evidence of genetic variants with obvious allelic effects on expression. I appreciate the authors’ efforts to increase the population diversity of study subjects in eQTL mapping studies and feel that they have made an important contribution to the eQTL literature.

The paper would still benefit from a thorough revision. My comments and concerns, detailed below, generally point to what I feel are lapses in precise language and careful presentation. I classify them all as “minor” comments. I hope that the authors find these comments constructive, helpful, and actionable.

Minor

1. In general, the paper would benefit from meticulous copyediting. There are several instances of grammatical errors or strange phrasing, particularly with articles (“a”, “an”, and “the”). See, e.g. lines 87 (“The eQTL mapping…” vs. “eQTL mapping”) or line 115 (“previous GWAS findings at SORT1 gene” should be “… at the SORT1 gene.”).

2. Normally, the subsection headers for results use titles that explain the scientific question to be answered in that section. Some of the headers, particularly lines 119, 230, and lines 246, state what was done but not what was learned.

3. The authors often state either explicitly (line 76) or implicitly (lines 93, 161, and Fig 3D) that AAs are an African population. While AAs are considered part of the 1000 Genomes AFR superpopulation, they are not exactly African: AAs are an admixed population with high average levels of African ancestry. Therefore, on line 93, where the authors state that “gene expression in Africans [is] unexplored.”, I encourage the authors to consider if statements like this in the paper are perhaps better phrased as something like “gene expression in populations with high levels of African ancestry,” akin to the language that appears in the manuscript cover letter.

4. Expanding on comment (2), the authors use LD maps from 1000 Genomes CEU and YRI populations in Fig 3D. As I understand it, the point is to argue that the LD structure in Africans differs from Europeans, and that this observation matters for the resolution at which eQTLs can be mapped in AAs. In that vein, it is curious that the authors did not use the 1000 Genomes ASW (African-Americans from the Southwest United States), since their study cohort is genetically closer to ASW than to either CEU or YRI. ASW could therefore provide a more comparable LD structure for the authors’ sample.

5. Line 107: The authors state that AAs have a large portion of AFR contribution to their genomes and a smaller EUR contribution. This is certainly intuitive to anybody who knows the history of AAs, but it is also an empirical observation of the genetics of AA individuals which the authors should justify. One paper that does this nicely is Baharian et al. (DOI: 10.1371/journal.pgen.1006059)

6. Line 110: The “first eQTL mapping in AA hepatocytes” is potentially contentious, since a casual reader may ask about the 15 AAs in the GTEx eQTL map of liver tissue. Presumably the human liver is not solely composed of hepatocytes, but this nuance is not mentioned here; instead, it appears in the Discussion on line 310. If there is particular utility in clean hepatocyte expression profiles uncontaminated with other liver cells, then the authors should state this in the Introduction and repeat it in the Discussion. In the meantime, it sounds more accurate for the authors to state that theirs is the “largest eQTL mapping to date in hepatocytes derived exclusively from AAs” or something similar. Adding the sample size (n = 60) at this juncture would help, along with a reason why the sample size may feel small when compared to the gargantuan modern GWAS sample sizes (e.g. deriving human hepatocytes requires actual human livers, which one can easily harvest in model organisms but not in human).

7. Line 127: I encourage the authors to cite a reference to KEGG here. There are numerous KEGG papers that fit the bill, but it should suffice to cite Kanehisa et al. 2016 (DOI: 10.1093/nar/gkv1070)

8. Line 134: Presumably a “lead eQTL” is the eQTL of largest effect for an eGene. The authors should clearly define what “lead eQTL” means here.

9. Fig 1E: the panel for TMEM116 expression shows that GTEx has only two samples with an allele dosage of 1 at the identified eQTL. The authors note this on lines 161-162 and assert that the SNP “is unique to African population[s]” and has a low MAF in GTEx. I have two minor qualms here. Firstly, the authors make an assertion without justification, which they can easily rectify by providing a (supplementary) figure showing the MAF of that variant in, say, all 1000 Genomes populations, akin to what Ensembl generates for variant lookups. Secondly, in Fig 1E, box-and-whisker plot for two data points is uninformative since the two points merely demarcate the extremes of the whiskers. The authors should consider if it is not more accurate to simply plot the dots for that case.

10. Fig 2: Why did the authors present panels 2A and 2B as box-and-whisker plots, but panels 2C and 2D as bar plot with error bars? It seems like the box-and-whisker plot is more informative here. Note that a similar question could be raised about the panels in S9 Fig. But S9 Fig shows adjacent bars with multiple (3+) colors, and a box-and-whisker plot with 3+ adjacent colors is probably harder to interpret than the current S9 Fig.

11. Fig 2 and Fig 3B: The bars and box-and-whisker plots presented here would be more complete if the authors added the data points underlying the summaries, as they did in Fig 1E and Fig 5A.

12. Lines 224-225: The authors speculate that the higher number of trait associations from overlapping eQTLs (those in GTEx and their AA cohort) vs. the AA-specific eQTLs could be due to “the biased representation of European cohorts in the GWAS catalog.” Could this not also occur with larger GWAS sample sizes? Is it known if genetic markers identified by GWAS in AA cohorts are more or less likely to associate with the AA-specific eQTLs? The authors’ assertion here would benefit from a more detailed explanation.

13. Fig 4: Is it possible to make the lines in these two panels color-blind friendly? They are already somewhat tricky to distinguish, and probably more so for readers with deuteranopia.

14. Line 251: Is there a particular reason for the choice of the FDR threshold 0.1 here? In other places in the paper (lines 409, 452, 674, 735) I see that the authors use an FDR threshold of 0.05. Do the results of the local ancestry eQTL mapping change for FDR-corrected p-values < 0.05?

15. Lines 200, 228, and 307: Following up on Comment (13), I am confused by these statements about FDR. FDR is normally stated as an adjustment procedure with an alpha level against which to compare adjusted p-values. Line 200 does not state what the FDR threshold is. Line 228 contains two statements of “FDR = x“, where I think the authors mean “FDR-corrected p-value = x”; again, the FDR threshold is not stated here. Line 307 clearly states “FDR = 0.143” which is an unusually high FDR threshold, and which I think is also referring to an FDR-adjusted p-value; a p-value at 0.143 is certainly not significant at FDR < 0.05 or 0.1, for example, and explains the authors’ statement about rs9923231.

16. Lines 291-292: the authors state, “It is possible that eQTLs related to drug response are discoverable only under specific contexts” as a plausible explanation for why so few eQTLs appear ontologically related to drug response. But a simpler possibility is that most researchers simply don’t have widespread access to well-designed eQTL studies of drug response. This is an important gap in the scientific literature, and the authors have license to remind readers about the pressing need for these kinds of studies.

17. Lines 303-304: The authors state that the sample size (n = 60) is a limitation of the study, and “type I and type II error may exists” [sic]. This is an odd choice of words, and perhaps a bit unfair to the authors, since the possibility of false positives and false negatives will exist so long as the FDR threshold is greater than 0! Certainly, a greater AA sample size may discover more AA-specific eQTLs, or it could also replicate many eQTLs already in GTEx as well but with a smaller effect size in AAs. No rational geneticist would argue that this study paints a complete picture of the genetic architecture of gene expression in AAs. Perhaps the authors can rephrase this statement as an admission of the limited statistical power available in their study, and that the interpretability of their results should be tempered accordingly. At the same time, the authors have demonstrated markers that show clear allelic effects on expression, so I feel that they should not sell themselves short here.

18. Lines 307-309: The authors point out that GTEx harvested postmortem tissue, and that this may introduce heterogeneity when comparing against their hepatocyte cell lines. There is some literature on how gene expression from postmortem tissue differs from that in living tissue samples, which I encourage the authors to discuss and cite; see, e.g. Ferreira et al. 2018 (DOI 10.1038/s41467-017-02772-x), Tolbert et al. 2018 (DOI: 10.1016/j.gene.2018.06.090), and Zhu et al. 2017 (DOI: 10.1038/s41598-017-05882-0) However, in Line 324, the authors state that their hepatocyte cell lines derive from non-transplantable organs. Since humans cannot survive without a liver, it seems reasonable to speculate that these hepatocytes could derive from deceased donors. Is this the case? Can the non-transplantable nature of the organ also affect gene expression? If not, then the authors should clearly state so, and justify how their hepatocyte cell lines are indeed different from postmortem cell lines.

19. Lines 366-368: The authors state that they discarded three samples based on PC plots of gene expression. I encourage the authors to add these plots as a supplementary figure, along with one that shows what constitutes a “normal” gene expression pattern in their sample.

20. Line 400: “… 6,453,712 SNPs” is more readable than the same number with no comma delimiters.

21. Line 415: If the authors refer to centimorgans here, then I think that “cm” should be “cM”.

22. References: Reference #37 is the preprint version of Reference #15. Reference #54 is a repeat of Reference #52.

23. Line 768, Fig 3 caption: in fairness to the EBI, which is a co-sponsor of the GWAS Catalog, perhaps the authors should write the “NHGRI/EBI GWAS Catalog”.

24. S2 Fig: The covariate names appear on the diagonal of this multipanel plot. This text on the diagonal is unnecessary since those covariate names appear as column and row names. Furthermore, the text is hard to read. I suggest that the authors simply remove it.

25. S6 Fig: It is confusing to have AA-specific eQTLs described in the context of the “AA” label in this figure. At first glance, I thought that AA-specific eQTLs all had higher average MAF than GTEx eQTLs, whereas the intent of the figure (I think) is to show that eQTLs estimated in the authors’ AA study cohort and replicated in GTEx have low MAF in GTEx. I encourage the authors to consider clarifying the labeling in this plot.

Reviewer #2: This is an important study to publish given the relative paucity of gene expression data in non-European and minority populations, and that liver QTL studies are particularly important for refining complex disease associations. Some of the findings of the paper underscores this importance (although I do outline below where I think some conclusions may be inappropriate). In general (with some exceptions) the authors seemed to have used robust data analysis pipelines, did appropriate statistical analyses, and generated good quality data. However, I do think some major revisions are necessary for publication. 

MAJOR CONCERNS

1. According to this now somewhat outdated review paper, https://www.ncbi.nlm.nih.gov/pmc/articles/PMC3418580/#B17, there are other eQTL liver studies, one of which includes African American subjects. Did the authors consider looking for resources outside of GTEx to compare their liver eQTLs with? Do they have a good rationale for only comparing their data only to GTEx, e.g. availability of RNASeq data, proper study design, etc.?

2. The GTEx sample size is ~ 3 folds larger than the new AA hepatocyte sample set. If the authors account for this sample size difference (by e.g. randomly selecting the same number of subjects from GTEx to identify eGenes and eQTLs, and repeating this a few times), how does the number of overlapping vs unique eGenes and eQTLs between the 2 data sets compare then? I.e. can you discover more with African ancestry eQTLs than European ancestry eQTLs if all else is equal? (Although point 4 below may make this somewhat difficult to compare directly)

3. On lines 204-229 the authors argue for increased fine-mapping utility because less AA hepatocyte eQTLs overlap with GWAS variants. The authors argue that this is due to decreased LD, but is it not also likely to be due to the smaller sample size of the AA hepatocyte data set? I suggest using the same framework as in 2 above to tease out the effect of sample size on the result set.

4. A major limitation to the conclusions of the study is the difference in liver tissue type in GTEx (noted by the authors in the Discussion, but perhaps not strongly enough). With the absence of additional supporting data, e.g. showing overall comparable gene expression patterns in GTEx vs AA. hepatocytes particularly for eGenes unique to AA hepatocytes, one cannot make strong claims about the additional discoveries being due to the African ancestry of the hepatocyte sample set (I don't think a higher MAF spectrum of the new eQTLs discovered is conclusive on its own). Either such additional supporting data should be added, or especially the abstract conclusions and author summary should be less strongly stated.

5. Lines 413-418: LDSC does not account for the long-range linkage disequilibrium patterns in admixed populations such as African Americans and these results can therefore be misleading. Either the results should be removed from the paper (as I don't think it is integral to the paper) or the analysis should be repeated using a more appropriate method e.g. https://www.biorxiv.org/content/biorxiv/early/2018/12/20/503144.full.pdf.

Minor

- Line 50: Preferably do not use the word biomarkers, as strong biomarkers for disease was not discovered by this study, and this may mislead people who read the author summary

- Line 57-58: Sentence does not make sense: "This is an important as fine-mapping of causative variants has relied heavily on function validation."

- Line 60: "gene drivers" - suggest rephrasing as "genes that drive disease"

- Line 76: "population" should not be singular

- The authors should very briefly define what an eGene is when first used, for readers who are not familiar with the term

- Lines 87-89 and elsewhere: "the" eQTL mapping: suggest dropping "the"

- Line 97: admixed populations (AAs)- should be admixed populations such as African Americans (AA)?

- Line 98: "for many" should be "of many"

- Line 106-107; "coming from" - suggest rephrasing "inherited from"?; suggest dropping "parental"

- Line 107: "This results" implies that it is admixture that is causing shorter blocks of LD in African Americans, where it is really the African ancestry that leads to shorter LD (whereas more genetic variation and different allele frequencies is a function of both African ancestry and admixture) - rephrase

- Line 115: suggest prefacing with "the" SORT1 gene

- Fig S8: The distributions on the plot look very similar. Perhaps use log scale or violin plots to more clearly show the differences.

- Line 173: "eQTLs groups" should be "eQTL group"

- Line 207: "Eight-one"?

- Line 218: "sufficient" should be "sufficiently"

- Line 219: "increase" should be "increased"

- Line 221: "potential" should be "potentially"

- Line 260: "heterogeneities" - as far as I know, heterogeneity does not have a plural form (but I may be wrong)

- Line 262: and additionally, less than 5% of subjects are non-European or non-Asian (https://www.nature.com/news/genomics-is-failing-on-diversity-1.20759)

- Lines 293-302: This was not described at all in results - move to results and briefly put in discussion?

- Line 340: but you would have no missing data with imputed data so why filter on missingness?

- Line 342: "the gene dosage" should be "gene dosages"

- Line 398: According to https://gtexportal.org/home/tissueSummaryPage, there are 208 liver samples with genotype data in GTEx - why the discrepancy in numbers?

- Line 404-405: This implies that opposite effect direction eQTLs were labelled AA-specific eQTLs - surely not?

- I suggest that at the very least, supporting Information files which report all significant eQTLs, exonQTLs, local ancestry QTLs for use in fine-mapping of other studies be provided (so researches don't have to create these data from scratch from dbGAP and GEO), and possibly even all summary statistics

Reviewer #3: Overview

In this work Zhong et al describe identification of eQTLs in hepatocytes measured in African American individuals. This topic is extremely important due to the historical underrepresentation of non-European individuals in molecular human genetic studies and thus likely to make a significant contribution to the field. I thought the manuscript was well-written and outlined clearly. With that said I have several major comments concerning the authors’ efforts

Major Comments

1. Throughout the manuscript the authors report the total number of significant expression-SNP associations (eQTLs) which is the result of a marginal association scan. This procedure is standard, however it is misleading to only report the total number and not the estimated independent number of eQTLs. For example, at a given gene if a causal eQTL exists (either directly in the data, or through tagging), then any other nearby SNP that is in LD should also exhibit a significant marginal effect (barring power to detect in the first place). I think it’s fine to report the total number of eQTLs found, but the number of independent signals should be emphasized as that is more descriptive of the underlying biology.

To remedy this, I encourage the authors to perform a conditional analysis on the lead eQTL to estimate residual regulatory action conditioned on the lead hit. This, coupled with the lead eQTLs, would paint a much clearer picture of the regulatory landscape for hepatocytes in AA individuals. As such, all downstream comparative analysis with EUR/GWAS catalog should prioritize using the lead and secondary SNPs over the complete set of associated SNPs.

This is made apparent when the authors report 236 eQTLs for SDHC and possible link to hepatocellular carcinoma. It is highly unlikely that this gene has 236 SNPs involved in regulation, compared to the more likely 1-2. If either the lead (or secondary eQTL if it exists) are not rs3935401 then the authors should remove this statement.

2. The authors report increased Fst at AA-specific eQTL SNPs compared to EUR-shared eQTL SNPs. From a purely descriptive standpoint I think this analysis is fine for emphasizing frequency differences between populations, but the typical interpretation of Fst is a consequence of selective pressure which I think would need much more work to show. Given that, the authors should clearly state the caveats of this interpterion of Fst, and state that one reason is differential tagging, and therefore power to detect, of the ungenotyped causals between populations.

3. Ascertaining on the shared and AA-specific eQTLs, the authors report LD score differentiation and how that may contribute to differentiation. Can the authors include the EUR LDscores at these same SNPs for a baseline comparison? If avg LDscore at the AA-specific eQTLs is lower than the LDscore for the same SNPs in EUR that would also support an argument regarding differential tagging of the causal eQTL.

4. It would be interesting to perform a cross-population fine-mapping of the eQTL results for both AA and EUR to characterize the shared landscape. For example, the authors argue that the fewer number of SNPs identified in the AA data enable more precise fine-mapping compared with the EUR GTEx dataset. This argument can be made much more precise by performing statistical fine-mapping using the AA dataset compared with the EUR GTEx dataset and quantifying the expected number of causal variants at each locus. This is computed from the average posterior probability at each locus under some fine-mapping software (e.g., FINEMAP, PAINTOR, SuSie, etc).

5. Given that African Americans have significant admixture on average, the authors separately perform a local-ancestry adjusted QTL analysis to identify eQTLs that are driven by ancestry differences. This is well-motivated, but I do not understand why the primary analysis did not include estimates of local ancestry for each gene to begin with. This is supported by their results of stronger enrichment for H3K27ac markers in the adjusted analysis compared with the un-adjusted. The manuscript could be improved significantly if the authors reported a single set of results first adjusting for local ancestry and then separately reporting which genes associated with local ancestry estimates.

**Have all data underlying the figures and results presented in the manuscript been provided?**

Reviewer #1: Yes

Reviewer #2: No: The authors list a GEO ID (GSE124076) but not a dbGAP ID for the genotype array data

Reviewer #3: Yes

PLOS authors have the option to publish the peer review history of their article (what does this mean?). If published, this will include your full peer review and any attached files.

Reviewer #1: Yes: Kevin L. Keys

Reviewer #2: Yes: Michelle Daya

Reviewer #3: No

---

## [Decision Letter · Decision Letter 1]

11 Feb 2020

Dear Dr Perera,

We are pleased to inform you that your manuscript entitled "Discovery of novel hepatocyte eQTLs in African Americans" has been editorially accepted for publication in PLOS Genetics. Congratulations!

Yours sincerely,

Christopher R Gignoux, Ph.D.

Guest Editor

PLOS Genetics

Scott Williams

Section Editor: Natural Variation

PLOS Genetics

Comments from the reviewers (if applicable):

Reviewer's Responses to Questions

**Comments to the Authors:**

Reviewer #1: The authors have clearly made a good-faith effort to address all of my concerns. I have no further concerns about the manuscript and thank the authors for their meticulous edits.

Reviewer #2: The authors have sufficiently addressed my comments.

Reviewer #3: The authors have performed a great deal of work addressing my comments and as a result the manuscript is significantly improved.

**Have all data underlying the figures and results presented in the manuscript been provided?**

Reviewer #1: Yes

Reviewer #2: Yes

Reviewer #3: Yes

PLOS authors have the option to publish the peer review history of their article (what does this mean?). If published, this will include your full peer review and any attached files.

Reviewer #1: Yes: Kevin L. Keys

Reviewer #2: Yes: Michelle Daya

Reviewer #3: No

**Data Deposition**

http://datadryad.org/submit?journalID=pgenetics&manu=PGENETICS-D-19-01205R1

**Press Queries**

---

## [Editor Report · Acceptance letter]

13 Apr 2020

PGENETICS-D-19-01205R1 

Discovery of novel hepatocyte eQTLs in African Americans 

Dear Dr Perera, 

We are pleased to inform you that your manuscript entitled "Discovery of novel hepatocyte eQTLs in African Americans" has been formally accepted for publication in PLOS Genetics! Your manuscript is now with our production department and you will be notified of the publication date in due course.

With kind regards,

Jason Norris

PLOS Genetics

On behalf of:
